# Enteral Nutrition Overview

**DOI:** 10.3390/nu14112180

**Published:** 2022-05-24

**Authors:** Jennifer Doley

**Affiliations:** Morrison Healthcare, Tucson, AZ 85745, USA; jenniferdoley@iammorrison.com

**Keywords:** enteral nutrition, formulas, aspiration, refeeding, COVID-19, proning, formula, blenderized formulas, feeding tubes

## Abstract

Enteral nutrition (EN) provides critical macro and micronutrients to individuals who cannot maintain sufficient oral intake to meet their nutritional needs. EN is most commonly required for neurological conditions that impair swallow function, such as stroke, amytrophic lateral sclerosis, and Parkinson’s disease. An inability to swallow due to mechanical ventilation and altered mental status are also common conditions that necessitate the use of EN. EN can be short or long term and delivered gastrically or post-pylorically. The expected duration and site of feeding determine the type of feeding tube used. Many commercial EN formulas are available. In addition to standard formulations, disease specific, peptide-based, and blenderized formulas are also available. Several other factors should be considered when providing EN, including timing and rate of initiation, advancement regimen, feeding modality, and risk of complications. Careful and comprehensive assessment of the patient will help to ensure that nutritionally complete and clinically appropriate EN is delivered safely.

## 1. Introduction

Enteral nutrition (EN) is necessary in individuals who are unable to maintain adequate nutritional intake by mouth. It is preferred to parenteral nutrition (PN) which is associated with a greater incidence of side effects such as hyperglycemia, electrolyte abnormalities, and infection rates, as well as more long-term complications such as PN associated liver disease and metabolic bone disease. EN is more physiologically natural and helps maintain gut integrity which supports immune function and protects against gut atrophy [1].

## 2. Indications/Contraindications

Inadequate intake may be caused by a variety of physiological or medical reasons. Perhaps the most common is dysphagia, which can be precipitated by chronic neurological conditions such as Parkinson’s disease or amyotrophic lateral sclerosis, as well as temporary or permanent dysfunction of the swallowing mechanism as a result of cerebrovascular accidents or conditions such as head and neck cancers. Inadequate intake may also result from a reduced level of consciousness or a significantly altered mental status due to dementia, mechanical ventilation, and hepatic or metabolic encephalopathy, among others. Inadequate oral intake may be caused by physiological symptoms related to diseases or their treatments, such as nausea, poor appetite, or taste changes. In these cases, the need for EN is usually temporary [1].

Inadequate intake may also be of a more prolonged and/or intermittent nature due to chronic conditions, in which case EN may be required on a long-term basis. These conditions include motility disorders such as gastroparesis and chronic intestinal pseudo-obstruction. While dietary interventions and medications are a first line of treatment for these conditions, EN may be necessary to maintain adequate nutritional status. In cases where EN is not tolerated, PN will be required [1].

EN is a recommended treatment for some individuals with short bowel syndrome (SBS), depending on the amount of bowel resected. While PN may be necessary to meet nutritional needs, EN can be a valuable therapy to aid in small bowel adaptation. Following a small bowel resection, the remaining intestine undergoes changes in structure and function that increase absorption. This adaptation can occur for up to two years after surgery. Some individuals with SBS may eventually be able to wean off PN [1].

Another condition in which EN can be used as a treatment to beneficially alter GI function is inflammatory bowel disease (IBD). Exclusive EN (or a liquid formula taken orally) has been shown to increase remission rates and reduce the need for steroids and surgeries. Benefits may not be seen for six to eight weeks and patient compliance can be challenging; however, exclusive EN may be a preferrable treatment when considering other treatment options such as surgery [2].

The primary contraindication for EN is a non-functional gastrointestinal (GI) tract. Conditions such as high output GI fistulas, bowel obstructions, paralytic or prolonged ileus, and mesenteric ischemia preclude the ability to provide EN and require parenteral nutrition (PN) in order to meet nutritional needs.

In cases where an individual cannot take nutrition orally and the GI tract cannot be accessed, PN may also be necessary [1]. In early critical illness, research has shown that there is no difference in clinical outcomes in patients that are fed EN vs. PN for short periods of time (5 to 7 days). Thus, PN may be an appropriate substitute should EN not be feasible in this population [3].

EN may be contraindicated in persons who are at the end of life and do not wish to receive aggressive medical and nutritional interventions. In those with advanced dementia, research has demonstrated that EN provision does not improve quality of life, mortality, and the incidence of pressure injuries [4,5]. In a position statement, the American Geriatrics Society specifically states that feeding tubes are not recommended in this population as their use is associated with increased agitation, use of restraints, and the development of new pressure injuries [6]. Decisions regarding nutrition support are ultimately up to the individual or their designated medical decision makers; however, clinicians must fully communicate the risks and potential outcomes of EN feeding so that informed decisions can be made prior to the placement of a feeding tube [1,4,5].

## 3. EN Administration

Components of EN administration include the timing of EN initiation and advancement, the site of feeding (i.e., gastric vs. enteric), the type of feeding tube, modality, and EN formula. Several factors influence these decisions, most notably medical conditions, risk of complications, and the expected duration of EN.

### 3.1. Timing

The ideal timing for the initiation of EN varies and is individual to the patient. EN should be initiated more aggressively for patients who are malnourished or at a high risk of malnutrition [1]. In critically ill patients specifically, early EN initiation within 24 to 48 h is recommended [7].

Other factors should be considered for critically ill patients who require EN. Hemodynamic instability is usually an indication to delay the initiation of EN. Enteral feeding while a patient is hypotensive may result in gut ischemia as the body prioritizes blood flow to more vital organs, such as the heart, brain, and lungs, leaving the GI tract with insufficient blood perfusion. Although this is a rare complication, the effects can be devastating, including necrotic bowel requiring resection and increased mortality. The American Society for Parenteral and Enteral Nutrition/Society of Critical Care Medicine (ASPEN/SCCM) guidelines for nutrition support in critically ill patients recommends that EN be withheld in the setting of hemodynamic instability, specifically a mean arterial pressure (MAP) of less than 60 mm Hg, multiple or escalating doses of vasopressors, or rising lactate levels [7]. If initiated in patients with tenuous hemodynamic stability, EN should be cautiously started at low rates of 10 to 20 mL/h and tolerance monitored closely. It should be noted, however, that the quality of evidence for this guideline is rated as very low [7]. Ultimately, clinical judgement should be used in decisions regarding the initiation of EN in consideration of the patient’s overall clinical condition and ongoing medical treatments. The European Society for Clinical Nutrition and Metabolism (ESPEN) guidelines for nutrition support in the intensive care unit (ICU) also suggest delayed initiation of EN in patients with hemodynamic instability, although a specific recommendation regarding MAP is not identified [8].

Although most research has been conducted on critically ill patients, early EN initiation has been studied in other conditions. Some studies have shown that head/neck cancer patients in whom feeding tubes were placed prior to the start of cancer treatments had overall better outcomes, including the prevention of weight loss, reduced hospital admissions, and an improved quality of life, compared to those who did not [9,10,11,12,13,14]. Early EN in patients with acute severe pancreatitis is associated with lower mortality, a lower incidence of multiple organ failure, and a reduced need for operative interventions [15,16]. In burn patients, EN as early as within four hours of injury has been associated with fewer complications, including infection rates [17]. ASPEN/SCCM guidelines recommend EN initiation within 4–6 h of injury for burn patients, although the authors acknowledge there are significant logistical barriers to achieving this goal [7].

### 3.2. EN Volume/Energy Goals

Research suggests that most patients, even those that are critically ill, can tolerate EN initiated at goal rate, although the goal rate may be quite low in patients receiving significant amounts of energy from other sources, such as lipid and dextrose based intravenous (IV) solutions [7]. Further, current guidelines suggest that there is no difference in the outcomes for low-malnutrition-risk critically ill patients that are fed trophic EN (usually defined as 10–20 mL/h), compared to full EN in the first seven days of hospitalization [3,7]. However, there is a paucity of data on this topic. Thus, clinicians should use their judgement in determining the goals for EN volume/energy provision. It should be noted that this issue is in reference to energy intake only; adequate protein, usually defined as 1.2 to 2 gm/kg in critically ill patients, should still be provided [3,7].

EN should be advanced more slowly if there is concern for refeeding syndrome, electrolyte disturbances, GI intolerance, hemodynamic instability, hyperglycemia, or hypercapnia. If not started at goal rate, EN can be initiated at 20 to 40 mL/h and advanced by 20–30 mL/h every 4–12 h, again depending on clinical judgement in consideration of the above factors [7].

#### 3.2.1. Refeeding Syndrome

Patients at a high risk for refeeding are those with malnutrition or little/no nutritional intake for a prolonged period. In refeeding, dextrose is rapidly acquired by the cells, resulting in extracellular to intracellular electrolyte shifts. Thiamin may also become depleted as it is a necessary cofactor for glucose-related metabolic processes. The hallmark of refeeding is hypophosphatemia, although hypokalemia and hypomagnesemia are also common. Table 1 details the ASPEN criteria for moderate and high risk for refeeding. The EN rate should be advanced more slowly. Deficient electrolytes should be replaced prior to starting EN and checked and replaced daily as needed until stable. Supplemental thiamin should also be administered. The ASPEN recommendations for the prevention and treatment of refeeding are more comprehensively detailed in Table 2 [18].

#### 3.2.2. Volume-Based Feeding

In volume-based feeding (VBF), a total daily goal volume of EN formula is determined and a goal rate calculated based on the number of hours that EN is scheduled to be infused. However, if EN is unexpectedly held for any reason the rate can be temporarily increased to make up for the time the EN was held. VBF charts are available to assist the nurse in calculating “make up” rates. Studies have shown that VBF protocols can result in an increase in the amount of formula received, compared to rate-based feeding regimens, and no difference in tolerance has been noted between the two methods [7,19].

### 3.3. Feeding Site

EN feeding sites are either gastric or post pyloric; post pyloric sites are duodenal or jejunal (past the ligament of Treitz). The ideal site for feeding is dependent on several factors, including the expected EN modality, risk for aspiration, and specific medical conditions. Most patients, even those that are critically ill, can tolerate EN that is fed gastrically [7,8]. Patients at a high risk of aspiration should be fed post-pylorically [7,8]. Aspiration risk factors include [7]:Inability to protect the airwayPresence of a nasoenteric tubeMechanical ventilationAge > 70 yearsPoor oral careInadequate nurse:patient ratioSupine position (head of bed not elevated)Neurological deficitsGastroesophageal refluxTransport out of the ICUUse of a bolus EN regimen

Impaired gastric motility that is unresponsive to prokinetic medications may also necessitate post-pyloric feeding [7].

Jejunal feedings may be preferred in patients with severe acute pancreatitis to avoid stimulation of the pancreas and gallbladder. However, recent research has indicated that gastric feeding may be just as well tolerated and safe with no differences seen in mortality, infectious complications, and the need for surgical intervention between gastric and jejunal feeding [20,21,22].

### 3.4. Modality

Modality of feeding is most dependent on expected tolerance and the feeding site; however, it is also influenced by the anticipated duration of EN and the availability of feeding equipment. In the ICU, patients are typically fed continuously. ESPEN guidelines recommend the exclusive use of continuous feeding in the ICU; however, ASPEN/SCCM guidelines do not [7,8]. If feeding is post-pyloric, EN should be continuous as boluses into the small intestine may result in abdominal discomfort and dumping syndrome [1].

#### 3.4.1. Continuous

In non-critically ill hospitalized patients, EN may be bolused if the feeding tube is placed gastrically and the patient has demonstrated a tolerance for continuous feeding. Some hospitalized patients may be started on bolus feeding if aspiration risk is low and there are no medical conditions such as gastric dysmotility that would cause concern for GI tolerance [1].

#### 3.4.2. Bolus

Bolus feeding is preferred in the home/community setting. Continuous feeding requires the use of a feeding pump which the patient must remain connected to for the duration of feeding. This may more significantly impede the activities of daily living and quality of life for individuals on continuous EN, compared to those on bolus EN regimens [1]. Further, feeding pump rental or purchase costs may be inhibitory if not covered by the patient’s health insurance.

#### 3.4.3. Intermittent

Intermittent feeding is continuous EN infusion that is held during certain time periods to dispense medications or undergo medical procedures. More commonly, however, intermittent EN is used as a strategy to promote oral intake in those that are transitioning to a diet. Nocturnal intermittent feedings are often utilized in these cases; holding EN during waking hours helps promote hunger, while providing EN at night ensures that the patient still receives adequate nutrition. If continuous feeding is necessary, nocturnal feeds are also preferred in the home setting to liberate the patient from the feeding pump for regular periods of time [1].

### 3.5. Feeding Tubes

The type of feeding tube is dictated by the site of feeding, i.e., gastric vs. post-pyloric. The expected duration of feeding, i.e., short term vs. long term, will also influence the choice of tube.

#### 3.5.1. Short Term

Short term feeding is defined as less than four to six weeks. Gastric and post-pyloric feeding tubes for short-term EN administration are inserted through the nares or may be placed orally in sedated mechanically ventilated patients. Post-pyloric feeding tube placement is generally more challenging than nasogastric tubes as post-pyloric tubes are more flexible and of a smaller bore size. Thus, they are more likely to become coiled in the stomach. There may also be a delay in starting EN due to the time it takes for the tip of the tube to migrate past the stomach and to conduct radiographic imaging to confirm its appropriate placement. However, patients that are awake may prefer post-pyloric tubes because they are more comfortable. These tubes may be a suitable alternative for patients that are reluctant to have a feeding tube placed, or those who express discomfort with nasogastric tubes. If post-pyloric feeding tubes are used for this reason, EN initiation need not be delayed until the tube is positioned post-pylorically. Dual lumen feeding tubes are also available for use in patients that need gastric decompression or suction while receiving EN via the jejunal port [1,23].

Patients with altered mental status may attempt to pull out nasally placed feeding tubes and sometimes succeed in doing so even when restrained. Devices called nasal bridles can be placed through the nasal septum, which secures the feeding tube in place to prevent removal. Research has shown that nasal bridles are safe and effectively reduce the incidence of tube dislodgement [23,24,25].

#### 3.5.2. Long Term

Gastrostomy or jejunostomy tubes are used for long term EN (greater than four to six weeks). Jejunostomy tubes are usually placed by laparoscopic surgery or radiologically, while gastrostomy feeding tubes are placed endoscopically in a GI lab or at bedside in the ICU. However, in some cases gastric tubes may need to be placed surgically or radiologically due to anatomical challenges, such as extreme abdominal adiposity, the presence of a large hiatal hernia, or prior surgical alterations to the GI tract. Feeding tubes can also be placed surgically when a patient is already undergoing another surgical procedure and the need for a feeding tube is anticipated. Benefits of long-term over short-term tubes include a lower risk of aspiration, as the gastroesophageal sphincter is not held open by a tube; reduced risk of tube displacement; no nasal or sinus discomfort; and reduced risk of pressure injuries on the nares [1,23,26]. Dual lumen tubes placed endoscopically or surgically may also be used for long-term feeding in chronic GI motility disorders such as gastroparesis [1].

## 4. EN Formulas

EN formulas can be categorized as standard, peptide-based, immune modulating, disease specific, and blenderized. The composition of these formulas varies by manufacturer but shares similar characteristics. Clinicians should be aware of the strength of evidence for the benefit of using specialty formulas for specific diseases or clinical conditions [1,7].

### 4.1. Standard

Standard formulas have intact nutrients, usually with carbohydrates in the form of maltodextrin and corn syrup solids, protein as soy protein isolate or caseinates, and fats such as safflower, canola, or soybean oil. They are available in different concentrations, from 1 to 2 kcal/mL; more concentrated formulas may be appropriate for individuals with conditions which require fluid restrictions, such as heart failure and renal disease, among others. Some standard formulas have fiber, usually a combination of soluble and insoluble fiber, while others are fiber-free [1].

The ASPEN/SCCM guidelines for nutrition support in critical illness state that a standard formula with or without fiber may be tolerated by most ICU patients [7]. Fiber-containing formulas should be avoided in patients who are hemodynamically unstable and at risk for bowel ischemia. Additionally, fiber-containing formulas should not be routinely used to promote bowel regularity or prevent diarrhea in this patient population [7]. Although standard formulas can be used for most critically ill patients, critical illness results in elevated protein needs and standard formulas are generally not high enough in protein to meet those needs without the use of supplemental protein modular [1].

### 4.2. Peptide-Based

Peptide-based formulas, sometimes referred to as elemental or semi-elemental, may be used in patients with malabsorption or who have demonstrated GI intolerance of standard formulas. These formulas are easier to digest as protein is hydrolyzed into small chain peptides, and fat sources include medium chain triglycerides and/or fish oil structured lipids. Peptide-based formulas are generally high in protein; some may contain up to 25 to 35% of total calories from protein, and are therefore of benefit in critically ill patients, especially in those receiving additional calories from IV solutions which necessitates a lower EN rate to avoid overfeeding. Many peptide-based formulas also contain higher amounts of antioxidant micronutrients, such as vitamins C, D, E, and selenium; and the omega-3 fatty acids eicosapentaentoic acid (EPA) and docosahexaenoic acid (DHA) [1].

### 4.3. Immune Modulating

Some peptide-based formulas are considered to be immune-enhancing or immune-modulating. In addition to the added antioxidant components found in other peptide-based formulas, they contain arginine and glutamine. Research on the use of immune-modulating formulas has demonstrated improved outcomes, such as reduced infection rates, reduced hospital length of stay (LOS), and reduced duration of mechanical ventilation in trauma and surgical ICU patients, but this has not been demonstrated in other groups such as medical or cardiac ICU patients. The ASPEN/SCCM nutrition support guidelines in critical care recommend the consideration of immune-modulating formulas in post-operative SICU and traumatic brain injury patients only. They rate the quality of evidence for this recommendation as very low [7].

It is hypothesized that supplemental arginine can result in negative side effects for septic patients due to arginine’s role in nitric oxide production, which may worsen hemodynamic stability. However, research has not demonstrated any adverse events in septic patients that are fed these formulas, nor have they demonstrated any benefit. The ASPEN/SCCM guidelines do not recommend the use of arginine containing formulas in septic patients [7].

The ASPEN/SCCM guidelines suggest that patients not already on glutamine-containing formulas should not receive supplemental glutamine [7]. However, the ESPEN guidelines recommend supplemental glutamine for the first 10–15 days on EN in burn patients, and the first five days in trauma patients [8].

### 4.4. Disease Specific

The ASPEN/SCCM nutrition support in the critical illness guidelines do not recommend the routine use of disease-specific formulas in medical and surgical ICU patients [7]. However, their use in some subsets of patients may be of benefit. The ESPEN guidelines do not specifically address disease-specific formulas [8].

#### 4.4.1. Hyperglycemia

Formulas that are developed for use in patients with hyperglycemia are lower in carbohydrates, higher in fat (usually in the form of mono-unsaturated fatty acids), and contain fiber. Some studies have demonstrated that these formulas can improve blood glucose control, while others have not. Further, the methodology of some research has been called into question [27] and the heterogeneity of studies have also made it difficult to draw conclusions [28]. In a recent review, the authors concluded that there was insufficient evidence to promote the use of low carbohydrate formulas [29]. Long-term use of these formulas has also not been sufficiently studied; therefore, potential long-term side effects or complications are unknown.

If low carbohydrate formulas are used, there are several contraindications to consider. These formulas should be avoided in patients with delayed gastric emptying, fat malabsorption, or other conditions in which high fat intake is not recommended. They should also be avoided in some critically ill patients, namely those that are at risk for bowel ischemia or obstruction in which fiber intake is contraindicated [29].

#### 4.4.2. Hepatic Disease

Hepatic formulas are very low in protein (as low as 10% of total calories) with a modified amino acid profile, namely an increased ratio of branched chain to aromatic amino acids. However, there is no noted benefit for these formulas, therefore their use is not recommended [7,30]. Protein restriction can cause or worsen malnutrition and is thus not recommended for patients with end stage liver disease. Additionally, protein restriction is no longer recommended in patients with hepatic encephalopathy as it increases protein catabolism [30] and has been shown to worsen clinical outcomes [31]. Treatment of hepatic encephalopathy should be limited to medical/medication interventions only.

#### 4.4.3. Chronic Kidney Disease

Renal formulas are calorically dense with moderate amounts of protein, and are lower in potassium, phosphorus, and magnesium. Protein restrictions are not warranted in most patients with chronic kidney disease (CKD) on renal replacement therapy (RRT). Therefore, renal EN formulas are generally not indicated unless elevated electrolyte levels cannot be successfully treated with RRT or medications. In individuals with CKD 3–5 that are not on RRT, a moderate protein restriction (0.6 g/kg) has been shown to delay the need for the initiation of RRT [32,33], in which case the long-term use of a renal formula would be of benefit. However, in hospitalized patients, a protein restriction should only be considered in CKD patients that are not on RRT or critically ill [34].

#### 4.4.4. Acute Kidney Injury

Protein needs are significantly elevated in acute kidney injury (AKI). RRT, especially continuous RRT, increases protein needs further [34,35]. The use of renal EN formulas can make adequate provision of protein challenging, thus their use is not recommended in this patient population. Like patients with CKD, renal formulas may be warranted in AKI if significantly elevated electrolyte levels cannot be successfully treated with RRT or medications. Labs should be monitored closely and the renal EN formula should be discontinued as soon as possible if it does not meet protein needs.

### 4.5. Blenderized

The use of blenderized feeding formulas has increased as demand for more “natural” formulas has grown, especially for individuals on long-term EN [36]. Blenderized formulas are less likely to contain food allergens, may be organic and vegan, and contain naturally occurring antioxidants not found in traditional commercial EN formulas. Patients have anecdotally reported improved GI tolerance to these formulas, although there is as yet limited evidence from randomized controlled trials on the benefit of blenderized formulas in patients with a GI intolerance for standard formulas or those with specific disease states [36,37,38].

Blenderized EN can be commercially prepared or non-commercially prepared (i.e., “homemade”) formulas. If non-commercial formulas are used, the patient should demonstrate a tolerance for bolus feeds and have access to a high-quality blender, refrigeration, clean water, and food. The benefits of commercially prepared products include consistent viscosity, nutritional adequacy, and a lower risk of microbial contamination; however, the use of non-commercial blenderized formulas is still a viable option in the home setting as steps can be taken to mitigate potential issues [39].

A larger bore feeding tube of French size 14 or greater is ideal to prevent clogging. Non-commercial formulas should be carefully and thoroughly blended to avoid inconsistent viscosity which increases the risk of clogging the feeding tube. Strainers may also be used to ensure the formula is sufficiently blended and less viscous. A registered dietitian nutritionist should be consulted to provide general support with the nutrition support regimen, including assistance in developing “recipes” to ensure that homemade formulas are nutritionally complete in both macro and micronutrients. Safe food handling techniques and proper storage will reduce the risk of food-borne illness. The patient and/or family must demonstrate understanding of formula preparation and administration with special attention to safe food practices [39].

## 5. Monitoring EN Tolerance

Gastric residual volumes (GRVs) have traditionally been used to assess tolerance of EN, however research has demonstrated that elevated GRVs are not correlated with incidence of aspiration, longer hospital and ICU LOS, or longer duration of mechanical ventilation. If GRVs are checked, ASPEN/SCCM guidelines suggest holding EN only if GRVs are greater than 500 mL. Tolerance of EN should be assessed by monitoring the patient’s bowel function and abdomen. EN should be withheld in the event of emesis/regurgitation, or significant abdominal distention or pain [7].

While monitoring bowel function is critical, symptoms such as diarrhea, constipation, or lack of bowel sounds usually do not warrant delayed initiation or cessation of EN [7]. Although commonly blamed on EN, diarrhea can be precipitated by several factors, especially in hospitalized patients. These include the use of high osmolality medications, medications that contain sorbitol or other non-digestible sugar alcohols, antibiotics resulting in altered gut flora, and infectious agents such as *c. difficile*. GI conditions such as SBS, irritable bowel syndrome, and inflammatory bowel disease may also result in diarrhea [40]. Constipation is also common in hospitalized patients, due in part to limited mobility and the use of narcotic medications. Depending on the cause, diarrhea or constipation may be mitigated with the use of fiber containing formulas, which usually contain a mixture of both soluble and insoluble fiber [41]. However, the etiology of the diarrhea or constipation should be investigated before changing to a fiber-containing formula [40]. Easily digestible peptide-based formulas with MCTs may also be used if malabsorption or maldigestion is suspected [7].

In hospitalized patients, fluid status should also be regularly monitored via physical exam to identify poor skin turgor and dry mucous membranes; measurement of input and output (I/O); serum sodium, chloride, blood urea nitrogen, and creatinine; and urine specific gravity and osmolarity. Temperature should also be monitored as hyperthermia can increase insensible fluid losses [1].

Complications of EN, as well as potential causes and treatments, are described in Table 3.

## 6. Other Considerations

Other issues should be considered when making decisions regarding EN administration, especially for hospitalized patients. Most notably, patient positioning, such as proning mechanically ventilated patients, medication administration, and supply shortages can affect the clinician’s ability to meet a patient’s nutritional needs via EN.

### 6.1. Patient Position

Placing the patient in a prone position has shown benefit in acute respiratory distress syndrome (ARDS) as it more evenly distributes ventilation throughout the lungs. This strategy has been more commonly used since the emergence of critical illness due to COVID-19 infection, as respiratory failure in this patient population resembles ARDS [42]. However, further research is needed to fully assess the benefit of proning in COVID-19-related respiratory failure.

Prone positioning may pose challenges to providing an adequate volume of EN to meet estimated nutritional needs. A feeding tube cannot be placed while a patient is in the prone position and tubes may become dislodged when the patient is turned [42]. Some clinicians prefer to hold or infuse EN at a lower rate while the patient is proned and significantly increase the EN rate while the patient is supine. However, most regimens recommend proning for 16 h, in which case the patient will only be in the supine position for eight hours [43]. EN may also be held for 30–60 min before the patient is scheduled to be turned, further reducing the time that EN may be infused at a higher rate [43]. These issues make the ability to meet the patient’s nutritional needs challenging, and in some cases, supplemental PN may be necessary.

Although limiting EN infusion rates during proning may seem prudent in theory, research has indicated that with some precautions, EN can be safely infused at goal rate while the patient is proned. Receiving EN while proned is not associated with higher GRV, or increased incidence of pneumonia, regurgitation, and aspiration. Post-pyloric feeding is recommended, and the head of bed should be elevated 10–25 degrees while proned, if possible. Gastric motility agents may also be given to reduce the risk of reflux and aspiration of formula. The head of the bed should be elevated 30–45 degrees, if possible, for any patient in the supine position who is at risk for aspiration [44,45,46,47,48,49].

### 6.2. Medications

In patients who are unable to swallow, feeding tubes may be used for medication administration. This increases the risk of clogging the feeding tube; however, adequately flushing the tube with warm water before and after each medication can mitigate this risk. Care should be taken to ensure the medication, if not already in liquid form, is adequately crushed and diluted. Some cannot be crushed, such as extended release and enteric-coated medications, as crushing will alter the rate of medication absorption. The distal site of the feeding tube must also be considered; medications that require gastric acid for digestion or activation should not be administered through a post-pyloric tube. If in doubt, a pharmacist should be consulted to ensure that all medications are ordered and administered appropriately. It should be noted that post-pyloric feeding tubes have a smaller bore and are thus at a higher risk of clogging than gastric tubes [23].

The bioavailability of some medications can be reduced or enhanced by concurrent administration of EN. Current ASPEN safe practice recommendations for EN do not specify which medications may require temporary suspension of EN, so healthcare institutions should review the available literature on food/medication interactions and develop facility specific policies [23]. If held, EN should be restarted promptly to avoid caloric deficits and the goal rate should be recalculated based on the number of hours that the EN is scheduled to be infused.

Clinicians should be aware of any medications that provide additional nutrients, which may result in overfeeding if not considered when calculating the EN goal rate or volume. Propofol, a commonly used sedative medication used in the ICU on mechanically ventilated patients, is dispensed in a lipid-based IV solution, and thus can deliver a substantial amount of calories. Clevidipine, a calcium channel blocker used to reduce blood pressure is also dispensed in a lipid-based IV solution. These medications, if infused at a high rate, may necessitate a reduction in the EN rate, making adequate protein delivery challenging. In these cases, the use of protein modulars may be necessary. Intravenous dextrose infusions can also be a considerable source of calories in hospitalized patients; however, supplemental dextrose is generally not indicated for patients receiving EN. If IV fluid is still required after EN has reached goal rate, changing from a dextrose-based to a dextrose-free solution may be preferable.

### 6.3. Supply Shortages

Because of the surge in cases of COVID-19 critical illness, periodic shortages of EN formulas, tubing sets, and pumps have occurred due to increased use and stockpiling, as well as manufacturing, shipping, and other supply chain difficulties. Strategies can be used to mitigate these challenges.

EN supplies and formulas can be sourced outside of a healthcare facility’s usual vendor or distributor. Different levels of the supply chain should be investigated as potential sources of product. If not available through the usual distributor, products may be available for purchase directly from the manufacturer. Items in short supply should be reserved those patients who are most critically ill [50,51,52].

#### 6.3.1. EN Formulas

If formulas packaged for exclusive EN use (i.e., ready to hang (RTH) products) are in short supply, the same product packaged in cartons or cans may be substituted. EN formula selection should be made judiciously with the understanding that flexibility may be needed; high protein formulas may need to be reserved for patients who would most benefit from them or who have demonstrated intolerance to other products. Equivalent EN formulas from other manufacturers may also be considered [50,51,52].

#### 6.3.2. Tubing Sets

Most healthcare facilities use a “closed” system for continuous feedings, in which the tubing set spikes directly into the RTH EN container. If alternative brands of tubing sets are purchased the pumps for that brand will also need to be obtained, as tubing sets are usually not interchangeable between pump brands. Alternatively, tube sets utilized in “open” systems may be used; in open EN systems, the formula is decanted into one bag and water to flush the tube into another bag. Some tubing sets may only have a bag for the formula and none for water which requires the nurse to conduct the water flushes manually. An open system is not ideal as it increases both nursing time and the risk of formula microbial contamination; however, if closed system EN tubing sets are in short supply, it may be necessary. Typically, smaller volume cartons or cans of formula are used for open systems; however, if these are not available, a larger RTH bottle or bag can be opened and decanted into the feeding bag. Any unused formula should be labeled and refrigerated, and then discarded after it expires. The timeframe for expiration is defined by the EN formula manufacturer [1].

To preserve feeding tube sets, maximum EN hang times may be extended. Most RTH EN formulas can be hung for 48 h; however, manufacturers recommend a hang time for feeding tube sets of no longer than 24 h. If the conservation of tubes is needed, the feeding tube set hang time may be temporarily increased to 48 h to preserve supplies. Approval should be obtained from the hospital’s infection prevention and nursing departments before utilizing this strategy [52].

In the event that no tubing sets or pumps are available, feedings may be bolused; however, this should be a last resort strategy in critically ill patients as it is more time consuming for nurses and increases the risk of GI intolerance and aspiration in high-risk patients.

#### 6.3.3. Feeding Pumps

To reserve pumps for ICU patients, any gastrically fed, non-critically ill, low aspiration risk patient may receive bolus feedings. If there are concerns regarding tolerance, a conservative bolus regimen can be initiated and the volume and frequency of feedings increased gradually. Critically ill patients, those fed post-pylorically, or those who have demonstrated an intolerance for gastric feeds should be given priority for pumps [51]. Pumps may also be shared between patients via the use of intermittent feeding (i.e., using the same pump, patient A is fed 8 am to 8 pm and patient B is fed 8 pm to 8 am) [52]. EN feeding pumps may also be available for rent [51].

In the event of a pump shortage, continuous feeding may also be administered via gravity drip, in which formula is decanted into a bag and a roller clamp is placed on the tube to regulate the flow of formula. The total volume of formula delivered can be estimated by counting the drops; 15 drops in approximately 15 s is equivalent to 60 mL. Using this information, a goal drip rate can be determined [50,51,52].

## 7. Conclusions

Patients who cannot receive sufficient nutrition by the oral route often require EN to meet their nutritional needs. EN can be short or long term, depending on the diagnosis or condition resulting in their inability to take food orally. Several factors should be considered, including the timing of EN initiation, feeding site, type of tube, feeding modality, initiation rate, advancement regimen, formula, and risk of complications. Careful and comprehensive assessment of the patient helps to ensure that nutritionally complete and clinically appropriate EN is delivered safely.

## Figures and Tables

**Table 1 nutrients-14-02180-t001:** ASPEN criteria to identify risk of refeeding in adults [18].

Criterion	Moderate Risk (2 Criteria)	Severe Risk (1 Criterion)
BMI (kg/m^2^)	16–18.5	<16
Weight Loss	5% in 1 mo	7.5% in 3 mo or >10% in 6 mo
Energy Intake	None/negligible for 5–6 d OR <75% of estimated needs for >7 d during acute illness/injury OR <75% of estimated needs >mo	None/negligible for >7 d OR <50% of estimated needs for >5 d during acute illness/injury OR <50% of estimated needs for >1 mo
Pre-feeding Potassium, Phosphorus, Magnesium	Minimally low levels OR normal current levels with recent low levels necessitating minimal or single-dose supplementation	Moderately/significantly low levels OR Minimally low/normal levels with recent low levels necessitating significant or multiple-dose supplementation
Subcutaneous Fat	Moderate loss	Severe loss
Muscle Mass	Mild/moderate loss	Severe loss
High Risk Comorbidities *	Moderate disease severity	Severe disease severity

* including, but not limited to: acquired immune deficiency syndrome, chronic alcohol or drug use, dysphagia/esophageal dysmotility, eating disorder, failure to thrive/malnutrition, prolonged emesis, malabsorption, cancer, post bariatric surgery. Reprinted with permission from Ref. [18]. Copyright 2020 American Society for Parenteral and Enteral Nutrition.

**Table 2 nutrients-14-02180-t002:** Prevention and treatment of refeeding syndrome in adults [18].

Aspect of Care	Recommendations
Initiation of Feeding	Limit dextrose to 100–150 g/d (including dextrose from other sources) or 10–20 kcal/kg for 1st 24 h Delay initiation until low electrolyte levels are replaced/corrected
Advancement of Feeding	Advance by ⅓ of goal every 1 to 2 days Delay advancement until low electrolyte levels are replaced/corrected
Electrolytes—Phosphorus, Potassium, Magnesium	Check before feeding Monitor every 12 h for first 3 days for high-risk patients Use established standards to replace low levels If severely low or difficult to correct levels, reduce dextrose/calories by 50% and readvance by ⅓ of goal every 1 to 2 days
Thiamin	Give 100 mg thiamin before feeding in high-risk patients Give 100 mg/d thiamin for 5–7 days in patients with severe starvation, chronic alcohol abuse, or other condition which results in high risk of thiamin deficiency
Monitoring	Vital signs every 4 h for first 24 h Daily weights Daily intake/output

Adapted with permission from Ref. [18]. Copyright 2020 American Society for Parenteral and Enteral Nutrition.

**Table 3 nutrients-14-02180-t003:** Common causes and treatment for EN related complications [1,28,40].

Complication	Potential Causes	Treatments
Diarrhea	Medications including oral electrolyte solutions, liquid medications with sorbitol or magnesium, laxatives, antibiotics, proton pump inhibitors, prokinetics, lactulose, glucose lowering agents, NSAIDS Fecal impaction *c. difficile* infection Altered gut flora GI disease, including IBD, SBS, post bariatric surgery, pancreatic insufficiency, SIBO	Consider change EN formula: ○ Less concentrated ○ Fiber containing ○ Peptide-based Add supplemental fiber if low risk for GI ischemia or obstruction Consider anti-diarrheal medication if *c.difficile* infection ruled out Discontinue potential offending medications (if possible)
Constipation	Medications including narcotics, oral iron supplements, phenytoin, calcium channel blockers Dehydration/inadequate fluid intake Inadequate fiber intake Decreased physical activity	Consider change to fiber-containing formula or add supplemental fiber if low risk for GI ischemia or obstruction Increase physical activity if possible Maintain adequate hydration Consider use of stool softeners, laxatives, and/or enemas Discontinue potential offending medications (if possible)
Nausea/Vomiting	Medications including opiate analgesics, anticholinergics GI disease, including IBD, IBS, GERD, pancreatitis, delayed gastric emptying Post bariatric surgery or pancreaticoduodenectomy	Consider low fat, low-fiber formula to avoid delayed gastric emptying Temporarily reduce EN infusion rate Administer EN at room temperature Feed post pylorically, especially if delayed gastric emptying suspected If on bolus regimen, reduce volume of bolus or change to continuous Consider prokinetic, antinausea, and antiemetic medications Discontinue potential offending medications (if possible)
Hyperglycemia	Medications including steroids Acute infection Critical illness/trauma Diabetes Overfeeding calories Insufficient provision of DM medications	Goal blood glucose 140–180 mg/dL in hospitalized patients Avoid overfeeding ○ Use IC (or evidence-based formulas when IC is not available) to calculate energy needs ○ Frequently reevaluate needs based on changes in clinical status Adjust/add insulin and/or oral hypoglycemic medications, insulin drip if severe hyperglycemia (ICU only) Discontinue potential offending medications (if possible)
Clogged Feeding Tube	Insufficient water flushes Improper medication administration Frequent GRV checks Use of high-fiber formulas	Flush tube with a minimum of 15 mL water before and after each medication and EN bolus administered Change to a fiber-free formula Review all medications to determine if any may be changed to form that does not need to be administered via the feeding tube Ensure medications are administered appropriately: ○ Do not crush enteric coated ○ Crushed medications should be a fine powder and fully dissolved in water

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
