# Peer review of "Enteral Nutrition Overview"

_nutrients, 2022, doi:10.3390/nu14112180_

Round 1
Reviewer 1 Report
well written
Author Response
thank you for your feedback
Reviewer 2 Report
The Review Article ‘Enteral Nutrition Overview’ provides comprehensive descriptions of enteral nutrition, including indications, administration, formula options, monitoring, and other considerations for enteral nutrition. The review summarized established ASPEN, ESPEN, and related guidelines, essentially covering well-accepted practices related to enteral nutrition. The general scope of the review does not address a gap in knowledge, and similar review articles have been published (Nutr Clin Pract. 2015;30:634-651). However, the discussion on COVID-19 considerations and considerations for formula and pump shortages offer a needed update, providing a timely summary of enteral nutrition for clinicians.
The article is very well organized and logical. The scope of the review was reasonable, offering breadth on the topic of enteral nutrition. The review was supported predominately by existing guidelines (ASPEN, ESPEN, KDOQI). Where there was a lack of guidelines (blenderized tube feeding and COVID 19/prone related content), recent articles were used appropriately to support the content.
Overall, the content of the review is in line with current practice and evidence-based nutrition support. There are no significant concerns with the article, and below is a list of minor comments to consider.
- In table 2 ‘Prevention and Treatment of Refeeding Syndrome,’ please clarify that this information is for adults since the pediatric recommendations are different.
- In lines 191-192 there is a comment that patients prefer post-pyloric feeding tubes when awake. There is no reference for this statement, and to my understanding, this is not common knowledge amongst practicing RDNs. Also, I wonder what the intended actions are based on this statement. I.e., would the recommendation be to offer postpyloric feeding for comfort, or would the recommendation be to offer gastric feeding and find ways to improve the comfort of the EN? Please consider revising including a reference and/or offering a suggestion on how to respond in clinical practice.
- The format of table 3 was challenging because the bullets were centered. Consider a left justification so that the hierarchy of the bullets within the table is clear. A left justification may also work with the refeeding tables for consistency.
Author Response
1 - added
2 - I don't have a reference for this, it's more anecdotal knowledge; I added a few sentences how this is relevant / can be applied to practice.
3 - agree! I will request the editor modify the formatting of these tables
Author Response
thank you for your feedback - responses attached
This manuscript is a resubmission of an earlier submission. The following is a list of the peer review reports and author responses from that submission.